# Is Anosognosia for Left-Sided Hemiplegia Due to a Specific Self-Awareness Defect or to a Poorly Conscious Working Mode Typical of the Right Hemisphere?

**DOI:** 10.3390/bs13120964

**Published:** 2023-11-23

**Authors:** Guido Gainotti

**Affiliations:** Institute of Neurology, Università Cattolica del Sacro Cuore, 00168 Rome, Italy; guido.gainotti@unicatt.it

**Keywords:** self-awareness, anosognosia for hemiplegia, unawareness of unilateral neglect, unconscious processing, right hemisphere working mode

## Abstract

This review aimed to evaluate whether the association between ‘anosognosia for hemiplegia’ and lesions of the right hemisphere points to a special self-awareness role of the right side of the brain, or could instead be due to a working mode typical of the right hemisphere. This latter viewpoint is consistent with a recently proposed model of human brain asymmetries that assumes that language lateralization in the left hemisphere might have increased the left hemisphere’s level of consciousness and intentionality in comparison with the right hemisphere’s less conscious and more automatic functioning. To assess these alternatives, I tried to ascertain whether anosognosia is greater for left-sided hemiplegia than for other disorders provoked by right brain lesions, or whether unawareness prevails in tasks more clearly related to the disruption of the right hemisphere’s more automatic (and less conscious) functioning. Data consistent with the first alternative would support the existence of a specific link between anosognosia for hemiplegia and self-awareness, whereas data supporting the second option would confirm the model linking anosognosia to a poorly conscious working mode typical of the right hemisphere. Analysis results showed that the incidence of anosognosia of the highly automatic syndrome of unilateral neglect was greater than that concerning the unawareness of left hemiplegia, suggesting that anosognosia for left-sided hemiplegia might be due to the poorly conscious working mode typical of the right hemisphere.

## 1. Introduction

Self-awareness is a multicomponent, hierarchically organized construct based on different sources of knowledge (e.g., [1,2,3,4]). It is generally acknowledged that the lower-level components of human self-awareness could be based on attentional consciousness of bodily functions and social relations. These lower-level components could be shared with other animals, and, in particular, with phylogenetically advanced social animals, such as primates, elephants and dolphins [3,4,5]. On the other hand, some higher-level components, based on declarative, autobiographic memory, introspection and narrative functions could be considered typically human capabilities. Several investigations have therefore tried to clarify the structure and neural substrates of self-awareness; some of these studies focused on the potential role of brain laterality in humans.

Drawing on clinical data gathered from unilateral brain-damaged patients, some authors have proposed that the right hemisphere might provide a critical contribution to the construction of self-awareness. This suggestion was based on the observation that right hemisphere lesions often provoke disturbances of the Self (e.g., [6,7,8,9]), and in particular on results obtained studying the neural correlates of anosognosia (disease unawareness), because several investigations (e.g., [10,11]) have shown that unawareness of hemiplegia is usually associated with right brain damage. The first author to stress the links between right hemisphere lesions, anosognosia of hemiplegia and ‘neuropathologies of the Self’ was probably Feinberg [9], who suggested that right hemisphere lesions might provoke alteration in the regulation of self-boundaries, and thus play a critical role in the aetiology of the ‘neuropathologies of the Self’. A clear inconsistency existed, however, between the right laterality of lesions observed in patients with unawareness of hemiplegia and results of activation studies conducted on normal subjects engaged in self-referential tasks concerning aspects of their self-awareness. As a matter of fact, if unawareness of hemiplegia is usually provoked by right-sided lesions, self-referential tasks usually produce left-hemispheric activation. For instance, in a meta-analysis of the neural substrate of autobiographical memory, Svoboda et al. [12] found a left-lateralised network that included both cortical and subcortical structures. Morin [13,14] stressed this inconsistency between results of clinical and activation studies and labelled it “the self-awareness anosognosia paradox” [14]. To solve this paradox, he proposed that anosognosia is not specifically due to right hemisphere lesions and can encompass multiple other related processes, such as self-monitoring problems, comparisons of performance pre-/post-brain damage and episodic memory disorders. This review further develops this position to show that the right laterality of lesions often observed in ‘anosognosia for hemiplegia’ does not necessarily point to a special role of the right side of the brain in self-awareness, but could be more generically due to some aspect of the right hemisphere’s functional organization. This viewpoint is consistent with positions assuming that specific awareness/control mechanisms are present in each functional brain system [15,16] and with data showing that, in addition to anosognosia for hemiplegia, unawareness of other deficits, such as unilateral neglect in stroke patients (e.g., [17,18]) or behavioural disturbances in fronto-temporal dementia (e.g., [19,20]), are usually associated with right brain damage. Furthermore, this viewpoint is in keeping with a model of the human brain asymmetries I recently proposed [21,22], which assumes that left lateralization and the shaping influence of language might have increased the left hemisphere’s level of consciousness and intentionality, in comparison with the right hemisphere’s less conscious and more automatic functioning. This review provides a short description of the model and its components, which could be more relevant to the problem of hemispheric asymmetries in functional awareness. Next, I match predictions made using this model to empirical observations that compare ‘anosognosia for hemiplegia’ to other forms of disease unawareness and, in particular, to anosognosia for unilateral neglect. The rationale for this procedure is that if a specific link exists between anosognosia for hemiplegia and self-awareness, then anosognosia should be greater for this defect than for other anosognosic disorders provoked by right brain lesions. If, on the contrary, anosognosia is related to functional features of the right hemisphere (such as its more automatic and less conscious functioning), then anosognosia should prevail in disabilities (such as unilateral neglect) that are more clearly related to disruption of these more automatic (and less conscious) functional mechanisms.

## 2. The Model Assuming That the Absence of Language Might Reduce the Consciousness and Intentionality of Right Hemisphere Functions, and This Model’s Implications for the Links between Anosognosia for Hemiplegia and Self-Awareness Disorders

The hypothesis assuming that different types of cognitive activities might be influenced by the presence or absence of language draws on Jackendoff’s [23] observation that language can influence not only the content of thought, but also its structure. According to this principle, levels of intentionality and consciousness might be greater in the left hemisphere where language is lateralized. Support for this hypothesis comes from various sources. Searle [24] stated that much of human consciousness is structured by language, Piaget [25] observed that children’s consciousness is determined by their capacity to verbally communicate their thoughts, and Brandon [26] distinguished the ‘practical intentionality’, shared with animals from the properly human ‘discursive intentionality’ that requires the presence of language and concepts. These general theoretic positions are supported by clinical observations showing that patients with left hemispheric lesions are mainly impaired in the most conscious and propositional aspects of brain functions, whereas those with right brain damage show less conscious and more automatic patterns of impairment. This inter-hemispheric contrast is particularly clear when we consider the automatic/propositional dissociation in the most important syndromes resulting from lesions of the left and right hemispheres, namely aphasia and unilateral spatial neglect. The observation that aphasic patients show a dissociation between impaired ‘propositional’ and relatively spared ‘automatic speech’ was made by Jackson in his classical papers on this subject (e.g., [27]); this observation was systematically confirmed by other authorities (e.g., [28]). Furthermore, the right hemisphere’s role in the production of automatic speech has been suggested by other authors (e.g., [29,30]).

On the other hand, the opposite dissociation has been consistently reported in unilateral spatial neglect; that is, the most frequent and debilitating consequence of strokes affect the right hemisphere, because it has a high prevalence of negative impacts on daily living activities [31]. Indeed, several investigations have documented the existence of a clear dissociation between the ‘exogenous’ (reflex and automatic) and endogenous (controlled and conscious) forms of contralateral attention orienting in unilateral neglect syndrome. Some neuropsychological studies (e.g., [32,33,34]) have, in fact, shown that neglect of the left half space occurs not only because stimuli arising in the left side of space do not draw the patients’ attention, but also because their attention is automatically captured by stimuli located in the right side of space. A clear demonstration of this phenomenon was provided by Mark et al. [34], who administered two equivalent versions of a cancellation task to patients with neglect. In the experimental condition, patients cancelled lines by erasing them, whereas in the control condition they cancelled lines using a pen stroke. As patients systematically began their activity from the right page margin, in the ‘erasing’ condition stimuli vanished from the right side of the page, whereas in the control condition they persisted in the right half space. Left-sided omissions were significantly more frequent in the ‘drawing over’ than in the ‘erasing’ condition, showing that the presence of stimuli in the right half space automatically attracted the patients’ attention, increasing the severity of neglect. Furthermore, other authors (e.g., [35,36]) observed that standard forms of neglect rehabilitation are based on a verbally induced, conscious exploration of the left half of space, suggesting that the endogenous components of spatial attention orienting might be preferentially mediated by the left hemisphere. Investigations concerning the neural substrates of neglect and brain mechanisms subsuming endogenous and exogenous forms of spatial attention orienting (e.g., [37,38,39]) are also consistent with these neuropsychological data, because they have shown that a dorsal and a ventral circuit are involved in attentional orienting. The ‘dorsal’ fronto-parietal circuit (represented both in the right and left hemispheres) should be involved in endogenous forms of spatial attention orienting, whereas the more ‘ventral’ circuit (more represented in the right hemisphere) could be mainly involved in its exogenous forms, allowing detection of unexpected, but relevant, events. The different involvements of the right and left hemispheres in the automatic and propositional aspects of brain functions could influence their level of consciousness, because “automatic” mechanisms tend to be associated with less conscious processing and “intentional/controlled” mechanisms with more conscious processing [40].

However, in addition to investigations concerning unilateral neglect syndrome, other experimental and clinical data have confirmed that the right hemisphere is characterized by automatic and partly unconscious stimulus processing.

Clinical observations supporting this claim have been made studying disorders of familiar people recognition via face (‘prosopagnosia’, see [41] for survey) and voice (‘phonagnosia’, see [42] for survey) modalities. Indeed, experimental and clinical studies have shown that the right hemisphere plays the most important role in both face and voice processing in normal subjects (see [43,44,45,46,47,48]) and in face and voice recognition disorders of patients with unilateral brain lesions (see [49,50,51,52]). Studying the extension of the ‘fusiform face area’ (specialized for face recognition) in normal subjects, Kanwisher et al. [44,45] showed that this cortical region is more represented in the right than in the left hemisphere; a similar asymmetry was observed by Belin and Zatorre [46] and Maguinness et al. [48] when investigating the human brain’s voice-identity recognition mechanisms. Consistent with these data are results obtained by authors [41,42,49,50,51,52] who studied hemispheric asymmetries in the production of face and voice recognition disorders. Furthermore, other investigations have shown that the basic mechanism underlying both prosopagnosia (e.g., [52,53,54]) and phonagnosia (e.g., [52,55]) consists of an inability to feel/evaluate if the face (or the voice) of a known person is familiar or not. This defect results from the fact that familiarity feelings are automatically generated when an incoming face (or voice) is matched with a previously established face (or voice) representation. Generation of these automatic familiarity feelings is, consequently, necessary to retrieve corresponding representations. As with spatial attention orienting, we can, therefore, say that the right hemisphere’s leading role in familiar people recognition is substantially due to its greater involvement in the automatic generation of corresponding familiarity feelings. A last aspect of the right hemisphere’s function, which indicates that the right side of the brain is characterized by automatic and only partly conscious stimulus processing, is its prevalence in emotional functions (e.g., [56,57,58,59]). Several theorists (e.g., [60,61,62,63,64]) have claimed that the emotional and cognitive systems (mainly shaped by language) are the most important adaptive systems that allow the human organism to select responses more suitable to the challenges of a partially unpredictable environment. However, the aims of these complementary adaptive systems could be different, because the ‘emotional’ should be considered as a primitive emergency system, whereas the ‘cognitive’ should be viewed as a more evolved and controlled adaptive system. Both the analysis of sensory information and the selection of the most appropriate responses made by these two complementary adaptive systems should, therefore, be different. The emotional system’s global, rapid and unconscious analysis should simply evaluate if an external situation is dangerous or pleasant, whereas the (slower) ‘cognitive’ analysis should be objective and exhaustive. Analogously, the emotional system should automatically select the most suitable response, choosing it from a small number of innate schemata, which correspond to the survival needs of the species, whereas the cognitive system should more slowly and consciously select a response more appropriate to the external situation. Also, the distinction between the left hemisphere’s ‘linguistic/cognitive organization’ and the right hemisphere’s ‘emotional organization’ indicates that the left side of the brain functions more consciously, and the right side of the brain functions less consciously and more automatically.

Partly related to the right hemisphere’s leading role in emotional functions is the final evidence supporting the assumption of its partly unconscious stimulus processing. This evidence concerns a line of research, triggered by Morris et al.’s discovery [65,66] of different (conscious vs. unconscious) processing of emotional stimuli of the left vs. right amygdala. In a first PET study, Morris et al. [65] showed that the unconscious processing (masked presentation) of a conditioned emotional face produced a significant neural response of the right (but not of the left) amygdala, whereas the opposite activation (i.e., a neural activity greater in the left than in the right amygdala) was provoked by the conscious processing (unmasked presentation) of the same stimulus. In a second paper, Morris et al. [66] tried to clarify this form of unconscious emotional learning’s mechanism, drawing on the distinction between a cortical and a subcortical pathway [67] that could reach the amygdala. During the masked (unconscious) presentation of conditioned emotional stimuli, they observed an increased correlation between activity in the right amygdala and its subcortical pathway (pulvinar and superior colliculus), whereas no masking-dependent changes in connectivity were found between the left amygdala and its subcortical pathway. Morris et al. [66] concluded that emotionally laden stimuli can be processed by the right hemisphere’s subcortical pathway without conscious awareness, whereas the left amygdala could play a greater role in the more complex and conscious processing of stimulus evaluation. These conclusions are supported by results of a meta-analysis of fMRI studies of amygdala responsivity to emotional stimuli [68], which confirmed a left lateralization for (conscious) verbal stimuli and a right lateralization for (unconscious) emotionally laden stimuli.

Taken together, results obtained from these clinical and experimental lines of research confirm that the right hemisphere’s levels of intentionality and consciousness might have been reduced by the absence of language, suggesting that the unawareness of functional disorders observed in patients with right brain lesions (e.g., [10,11,16,17,18,19,20]) might, at least in part, be due to this aspect of the right hemisphere’s functional organization.

## 3. An Attempt to Match Predictions Based on the Model with Data Reported in the Literature Regarding Anosognosia for Hemiplegia and Other Disorders Provoked by Right Brain Lesions

Data discussed in the previous section suggested that the general unawareness of functional disorders observed in patients with right brain lesions might be due to the right hemisphere’s lower levels of consciousness and intentionality, but did not clarify if a specific link exists between anosognosia for hemiplegia and self-awareness. To address this issue, I tried to evaluate if disease unawareness is greater for hemiplegia (which concerns body-related components of the Self) than for other disorders not related to the Self, but also provoked by right brain lesions. There were two main reasons to compare ‘anosognosia for left-sided hemiplegia’ with ‘anosognosia for left-sided spatial neglect’. The first reason was based on their similarities; both forms of unawareness are observed in patients with extended lesions in the right sylvian artery area, and severely impact daily living activities. The second reason was based on their differences; anosognosia for hemiplegia can be related to the Self, whereas anosognosia for left-sided spatial neglect is unrelated to this construct. To match the level of disease unawareness observed in these two conditions, I comprehensively (although not systematically) reviewed investigations that had assessed their incidence and severity in many subjects. 

### 3.1. Data Review Methodology concerning Anosognosia for Hemiplegia and Unilateral Neglect and Difficulties Encountered during This Comparative Survey

To compare the level of disease unawareness observed in these two conditions, I reviewed relevant published data using PubMed to search ad hoc studies via diagnostic keywords (“anosognosia of hemiplegia” and “anosognosia of unilateral neglect”) and focused on frequently cited papers that included sizeable patient samples. However, some methodological difficulties emerged after this comparative survey began. The first difficulty encountered using these search criteria was a numerical discrepancy between the large number of studies that had assessed the frequency of anosognosia for hemiplegia and the small number of studies that had taken into account the frequency of unawareness of unilateral spatial neglect. I identified 15 studies [10,11,69,70,71,72,73,74,75,76,77,78,79,80,81] that had investigated anosognosia for hemiplegia and 7 [17,18,82,83,84,85,86] that had assessed anosognosia for extrapersonal neglect for a large number of patients. This difference was probably due the fact that it is much simpler to clinically detect a discrepancy between the objective and the subjective evaluation of hemiplegia than it is to detect a similar discrepancy in the evaluation of unilateral neglect. Hemiplegia’s severity is immediately evident to the examiner, whereas the presence and severity of neglect can only be documented by specific neuropsychological tasks. The evaluation of unawareness of unilateral neglect therefore requires much more complex experimental designs than those required to detect the presence of anosognosia for hemiplegia. This interpretation was confirmed by the observation that most studies that investigated the frequency of anosognosia for hemiplegia in a large number of right and left brain-damaged patients were published in the second half of the last century, whereas most studies that assessed the frequency of unawareness of unilateral neglect were published much more recently. For instance, the three investigations that studied the incidence of anosognosia for hemiplegia in the highest number of unilateral brain-damaged patients were published in 1952 by Nathanson et al. [69], in 1976 by Green and Hamilton [70] and in 1978 by Cutting [71], whereas the study that investigated the highest number of patients with anosognosia for unilateral neglect was published in 2023 by Karataş et al. [86]. A second methodological difficulty was that even if lesions due to stroke were most frequently reported in studies of anosognosia for hemiplegia and of unawareness of unilateral neglect, these lesions and their localization could be heterogeneous. A different lesion pattern could therefore lead to anosognosia for hemiplegia and for neglect in the right hemisphere. 

A third methodological difficulty was that clinical studies reported anosognosia for hemiplegia frequencies that ranged from less than 20% (e.g., [76,77,79]) to more than 50% (e.g., [71]). This variability is probably related to several factors, including the time elapsed since a patient’s stroke, because a progressive recovery from anosognosia is often observed within the first 3 months following stroke (e.g., [71,75,77,80]). Some authors (e.g., [78]) have shown that an increase in time elapsed since brain injury can lead to a significant reduction (from 44 to 20%) in the incidence of anosognosia. Other variables could include the criterion used to assess anosognosia for motor impairment and the different settings (such as acute or rehabilitation hospitals or the community) in which studies were conducted. Some authors noted that the incidence of anosognosia for hemiplegia could vary even within apparently homogeneous settings; according to Orfei et al. [10], it varies from 8 to 34% in acute hospital studies.

The final methodological problem was that the aim of some reviewed investigations was not to evaluate the incidence of these forms of anosognosia in patients with lesions of the right or left hemisphere, but to evaluate the impact of these types of disease unawareness on functional outcome or discharge from the hospital (e.g., [76,80,85,86]). Therefore, such studies did not take into account the relationship between incidence of anosognosia and the side of brain in which the lesion occurred, but only the relationship between the presence of anosognosia and the functional outcome or discharge from the hospital within a given time limit.

### 3.2. Solution Chosen to Overcome These Methodological Difficulties

For all the above reasons, the most appropriate comparison for evaluating whether disease unawareness is greater for hemiplegia than for unilateral neglect was not matching results obtained in studies taking into account anosognosia for hemiplegia and for unilateral neglect separately, but rather comparing results obtained in studies in which anosognosia for hemiplegia and for unilateral neglect had been investigated in the same patients, and therefore obtained in the same settings and using similar criteria. 

For this reason, Table 1 reports data from results obtained in the five studies [17,18,77,82,83] found in the neuropsychological literature in which anosognosia for hemiplegia and for unilateral neglect had been investigated in the same patients. 

Results obtained in investigations [18,77,82,83] consistently showed that the incidence of unawareness of unilateral neglect was greater than that concerning the unawareness of left hemiplegia. Furthermore, results of [17] showed that right brain-damaged patients exhibiting a discrepancy between these two forms of anosognosia were usually unaware of their tendency to neglect stimuli lying on the left side of their extrapersonal space, but were aware of their severe motor defect.

## 4. Concluding Remarks

The aim of this investigation was to evaluate the observation that injury to the right hemisphere consistently produces a lack of awareness of hemiplegia and that this demonstrates the right hemisphere’s critical role in the construction of self-awareness. More specifically, I assumed that if anosognosia of hemiplegia was more frequent than anosognosia of unilateral neglect in right brain-damaged patients, this would indicate that the right hemisphere is related to the Self. In contrast, if anosognosia of unilateral neglect was more frequent than unawareness of hemiplegia (as this survey’s data suggest), then this finding could suggest either that anosognosia of hemiplegia is a by-product of unawareness of unilateral neglect, or that the right hemisphere has a less conscious working mode than the left hemisphere. Two of this survey’s findings are at odds with the hypothesis that the right hemisphere might provide a critical contribution to the construction of self-awareness, and rather suggest that anosognosia for hemiplegia might be an instance of the right hemisphere’s general tendency to be unaware of disorders provoked by its lesions. The first finding is that a strong relationship can be found not only between right hemisphere lesions and unawareness of hemiplegia (e.g., [10,11,69,70,71,72,73,74,75,76,77,78,79,80,81]), but also between right hemisphere lesions and anosognosia of other lateralized sensory defects [77,87,88] (e.g., unilateral spatial disorders (e.g., [17,18,82,83,84,85,86]) and emotional disorders (e.g., [19,20]). The second finding is that when the incidence of unawareness of hemiplegia and of unilateral neglect in studies that investigated these two forms of anosognosia in the same patients were compared, anosognosia was associated more with attentional rather than motor disorders. These results are clearly at odds with expectations based on the assumption that unawareness of hemiplegia should be considered a disturbance of the Self (e.g., [17,18,77,82,83]). 

If these survey results have clarified this aspect of this investigation, showing that unawareness of left hemiplegia cannot be considered as a disturbance of the Self, questions remain regarding the mechanism underlying the greater level of anosognosia shown by right brain-damaged patients regarding their neglect in comparison to their hemiplegia. According to the model on which I based this investigation, the right hemisphere (due to the left lateralization of language) could be less conscious and more automatic than the left hemisphere. This could explain the observed asymmetry between unawareness of neglect and of hemiplegia, because in neglect patients’ defective spatial attention orienting is due to a disruption of automatic mechanisms. I previously discussed (e.g., [21,22]) clinical data that support this model; reviews of other important clinical syndromes suggested that the right hemisphere and its pathology played a greater role in various disorders caused by the disruption of automatic functions. The present observations could, therefore, be interpreted within the same context.

An alternative interpretation of the greater incidence of anosognosia for unilateral neglect than of unawareness of hemiplegia in right brain-damaged patients could consist of the assumption that in some patients the anosognosia of left-sided hemiplegia might simply be the byproduct of severe ipsilateral neglect. In such cases, the automatic capture of attention by stimuli lying in the right (contralateral) half of the patient’s space could lead the patient to ignore limbs lying in the neglected left side of their space (see [89,90] for different positions regarding this hypothesis) and their severe functional defect. In these patients, anosognosia of hemiplegia should be considered a disorder due to an extension of the neglect syndrome to the discovery of the left hemiplegia.

## Figures and Tables

**Table 1 behavsci-13-00964-t001:** Investigations that assessed anosognosia for hemiplegia and unilateral visual neglect in the same patients.

[17] Jehkonen et al. (2000) assessed 57 patients with acute right hemisphere infarction; if a patient showed a double dissociation between anosognosia for hemiparesis and for unilateral neglect, this indicated that anosognosias are specific and independent impairments of awareness. They observed a double dissociation in 12 patients; however, 10 patients were unaware of neglect, but were aware of hemiparesis, whereas only 2 patients were unaware of hemiparesis, but were aware of neglect. [18] Ronchi et al. (2014) investigated unawareness of different aspects of unilateral neglect and linguistic and motor tasks in 29 right brain-damaged patients (17 with left spatial neglect). Patients were less aware of the attentional than the motor defect, because a clinical form of anosognosia for unilateral neglect was found in 17/17 (100%) neglect patients, whereas all right brain-damaged patients, as a group, were aware of their motor impairments. [77] Marcel et al. (2004) studied awareness of motor and sensory defects and of unilateral neglect in 62 stroke patients (42 with right-sided lesions and 22 with left-sided lesions). In right brain-damaged patients, anosognosia for hemiparesis (12/42 = 29%) was less frequent than unawareness of neglect (8/16 = 50%). [82] Bisiach et al. (1986) assessed anosognosia for hemiplegia and for hemianopia/unilateral neglect in 36 acute right brain-damaged patients. Anosognosia for hemiplegia (12/36 patients = 33%) was again less frequent than unawareness of visual defects (28/32 patients = 88%). [83] Berti et al. (1996) studied 34 right brain-damaged patients with complete hemiplegia (27 of whom had neglect dyslexia or drawing neglect). Patients were less aware of their neglect than of their severe motor deficit; anosognosia for neglect dyslexia was shown by 6 of 10 patients (60%), anosognosia for drawing neglect by 8 of 17 patients (47%), and anosognosia for hemiplegia by 9 of 34 patients (26.5%).

## Data Availability

Not applicable.

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
