# Peer review of "Is Anosognosia for Left-Sided Hemiplegia Due to a Specific Self-Awareness Defect or to a Poorly Conscious Working Mode Typical of the Right Hemisphere?"

_behavsci, 2023, doi:10.3390/bs13120964_

Round 1

Reviewer 1 Report

Comments and Suggestions for Authors

In the manuscript submitted to Behavioral Sciences Gainotti reviews the literature on the clinical combination of left-sided hemiplegia with anosognosia and compares it to the incidence of anosognosia and spatial neglect. The aim was to establish whether hemiplegia, which supposedly is related more closely to self-awareness than spatial neglect, might less frequently lead to anosognosia than neglect. To this end, a literature search was performed. Even though, the question is intriguing and might possibly connect to larger themes about the role of the two hemispheres in consciousness, ultimately the methodology and presentation in this manuscript are not strong enough to come to any definite conclusions.

Regarding the methodology, it is never clearly stated how the literature search was conducted (search terms, databases, time frame) and the studies were selected. If this were a narrative review, an unsystematic approach would be acceptable, but seeing as the incidence of a symptom combination is being investigated and stated in the concluding remarks, more details are required in my view.

 Additionally, from the text I find it difficult to understand, what kind of cerebral lesions were the cause of the clinical symptoms described. Since the lesion patterns leading to hemiplegia, neglect and anosognosia in the right hemisphere may differ, the studies might be difficult to compare. Since neglect is an additional symptom due to damage to a network between the ventrolateral frontal lobe, the upper and medium temporal lobe and the inferior parietal lobe, it might not be surprising that anosognosia is more frequently encountered than in presumably smaller lesions with isolated hemiplegia (see (Pedersen et al., 1996; Hartman-Maeir et al., 2003).

Moreover, if the overarching theme is the hypothesis that the left hemisphere has a more conscious working mode than the right, would it not be expedient to compare rates of anosognosia after injury to right and left hemisphere?

Lines 114-120: I believe this describes the extinction phenomenon, which can occur in combination or separately to neglect, and does not differentiate well between “endogenous” and “exogenous” forms of neglect (see Becker et al, “Neuroimaging of exe position reveals spatial neglect”, Brain 2010, Karnath et al “The anatomy of spatial neglect”, Neuropsychologia 2012). 

Importantly, table 1 appears not to be very clearly structured and incorporates multiple sentences instead of well-defined categories.

In summary, even though it is not supposed to be a systematic review, I believe the manuscript in its current form does not achieve to scientifically answer the question in the title and would therefore recommend publication only after major revision.

Comments on the Quality of English Language

Regarding English language and style, there are several orthographic mistakes errors (e.g. line 240: “discrepancy” not “dyscrepancy”, line 288: “literature” not “litterature”, line 254: “century” not “centusy”, line 320: “consists” not “consiste”). 

Reviewer 2 Report

Comments and Suggestions for Authors

Thank you very much for letting me review this article. It is an interesting and well-written review article that evaluates if the association between ‘anosognosia for hemiplegia’ and right hemisphere lesions points to a role of right hemisphere of the brain in self-awareness. This review has sufficient details and covers the findings till date. However, there are few typographical and spelling errors that needs to be corrected e.g. line 20 “to” repetition, line 58 “between”, line 153 “in” repeated etc.

Reviewer 3 Report

Comments and Suggestions for Authors

The manuscript entitled “Is anosognosia for left-sided hemiplegia due to a specific defect of Self awareness or to a poorly conscious working mode typical of the right hemisphere?” by Guido Gainotti. evaluated if the association between ‘anosognosia for hemiplegia and lesions of the right hemisphere points to a special role of this side of the brain in self-awareness, or could be more generically due to some working mode typical of the right hemisphere.

This is a weak review and hard to follow with several grammatical and typing mistakes. Also there is no real table 1 and it is more of a retelling of what previous work found without novelty. Moreover, this review contains personal opinions without a solid scientific basis to support these opinions. There is a lack of consistency regarding the outline of subsections. In general, the lack of detailed and relevant recent information dampens enthusiasm for this work.

Comments on the Quality of English Language

Hard to follow with several grammatical and typing mistakes

Round 2

Reviewer 1 Report

Comments and Suggestions for Authors

Interestingly, this is one of the few scientific papers that points towards an abundance in previous literature ("more than 300 papers") as an obstacle to systematic analysis. As stated in my first review, it is not an acceptable strategy to compare different etiologies and locations of right hemispheric lesions to each other. Therefore, selecting only a few and heterogenous studies is not a scientifically valid method to prove a hypothesis. Additionally, table 1 still has no discernible structure and does not adequately display results from a literature search. Numerous persistent typographic errors are another nuisance. 

Actually, I do agree with the sensible addition of the last paragraph stating that an alternative explanation for the coincidence of neglect and anosognosia might be due to reduced sensory input from the affected side. This, however, would provide an excellent antithesis to the proposed "poorly conscious working-mode" of the right hemisphere and should therefore be a much more prominent part of the discussion. 

All in all, I am afraid the presented work has not been sufficiently amended to justify publication in Behavioral Sciences. 

Reviewer 3 Report

Comments and Suggestions for Authors

Thanks for improving the manuscript. Table 1 still needs to be organized and show items for comparison.

Comments on the Quality of English Language

Needs improvements. 

Round 3

Reviewer 1 Report

Comments and Suggestions for Authors

As stated in my second review, I found insufficient improvement of the manuscript to justify publication. The third version does little to amend the unsystematic table 1, which is more of an enumeration, and furthermore contains no changes to methods or discussion. 

Comments on the Quality of English Language

Minor inconsistencies (e.g. simultaneous use of "lateralisation" and "lateralization") as well as typing errors ("futhermore", "litterature"). 
